# Hydroponic Cultivation of Medicinal Plants—Plant Organs and Hydroponic Systems: Techniques and Trends

Hallam R. Atherton [1] and Pomin Li [2,*]

1 Department of Tropical Agriculture and International Cooperation, National Pingtung University of Science and Technology, No. 1, Xuefu Rd., Neipu Township, Pingtung 912, Taiwan
2 Department of Biomechatronics Engineering, National Pingtung University of Science and Technology, No. 1, Xuefu Rd., Neipu Township, Pingtung 912, Taiwan
* Correspondence: pomin@mail.npust.edu.tw

**Abstract:** Medicinal plants are a globally important resource for the treatment of disease and improvement of human health, and the only form of healthcare used by millions of people. Currently, the status of many medicinal species is threatened by overharvesting caused by increasing demand. While many species have been cultivated in soil, the resulting material often contains lower levels of medicinal compounds than that of wild-harvested plants. Hydroponic cultivation of medicinal plants may provide sustainable access, with research demonstrating increased secondary metabolite content in some species compared to both wild and soil-grown plants. There are a variety of hydroponic systems and techniques available, each with its own advantages and disadvantages. As medicinal compounds are often localised within specific plant organs, selecting hydroponic systems that maximise the yield of the desired organs and metabolites is necessary. However, as of yet there has been no attempt to examine the literature with regards to the cultivation of specific plant organs of medicinal plants in hydroponics. This article explores the use of different hydroponic systems to cultivate medicinal plants and how the choice of system differs depending on the desired plant organ, as well as highlighting studies with novel outcomes that may provide value to the field.

**Keywords:** hydroponics; medicinal and aromatic plants; secondary metabolites; soilless cultivation; sustainability

## 1. Introduction

Plants produce a variety of different chemicals throughout the course of their growth. Some of these compounds, known as secondary metabolites, are beneficial to human health. They have a wide variety of uses, including reducing inflammation, treating diabetes, reducing cancer risk, preventing cardiovascular disease, and some have antimicrobial effects [1]. The production of pharmaceutical drugs relies on many of these chemicals. Up to 60% of anti-infectious or anti-tumour drugs available are sourced from nature [2], and approximately 50% of modern pharmaceuticals overall are natural [3]. Currently, there are around 70,000 different species of plants used by both modern and traditional medicinal systems globally [4]. These plants, known as medicinal plants, have been used for thousands of years to treat health problems. In traditional Chinese medicine (TCM), for example, 80% of medicines are sourced from plants [5]. Traditional medicines are of particular importance in rural areas as up to 90% of the world's rural population use traditional medicine, due to limited access to modern medical facilities [6]. In China, 40% of people rely exclusively on traditional medicines, while in Africa, the proportion is even greater at 80% [7]. Therefore, maintaining the availability of this valuable source of medicinal treatments is of great importance.

However, the increasing interest in medicinal plants also has negative consequences. A number of medicinal plants are listed as endangered species, and the overharvesting of them in the wild puts them at greater threat of extinction [5]. In addition, purveyors

of medicinal plants often, intentionally or unintentionally, sell plants that are incorrectly labelled, and samples vary in their secondary metabolite content or are contaminated with heavy metals [8]. The use of incorrect species in herbal medicines could result in an ineffective medicine or even adverse effects. Such instances have occurred where substitution, misidentification, and contamination have resulted in severe side effects and had the potential to cause fatalities [9]. Consequently, it is key that the environmental impact of using medicinal plants is reduced and that the safety and quality of medicinal material is maintained.

To ensure a stable supply of medicinal plants, efforts to cultivate them have been made. In China, around 200 species of medicinal herbs are cultivated on over 9.3 million hectares of land [10]. Similarly, research into the cultivation of Chinese medicinal plants has been conducted in Germany, and a selection of species have reached commercial production [11]. However, there are also disadvantages to cultivating medicinal plants in the field. The concentrations of secondary metabolites present in wild populations are often higher due to the environmental stresses and competitive pressure of other species, while their slower growth rates also result in the increased accumulation of active ingredients [12,13]. For example, in a review of studies examining the cultivation of *Hypericum perforatum* L., Bruni and Sacchetti [14] observed that cultivation studies conducted in different countries reported hypericin, pseudohypericin, and hyperforin concentrations that varied widely. Similarly, differences in the salidroside yield of *Rhodiola sachalinensis* Boriss. (synonymous of *Rhodiola rosea* L.) roots were observed in response to soil characteristics, such as organic matter, nitrogen, phosphorous, and potassium content, as well as pH [15]. Additionally, seasonal variations in temperature and moisture have also been attributed to differences in the levels of phenolic compounds and saponins in *Tulbaghia violacea* Harv., *Hypoxis hemerocallidea* Fisch. & C.A.Mey., *Drimia robusta* Baker (synonymous of *Drimia elata* Jacq. ex Willd.), and *Merwilla plumbea* (Lindl.) Speta plants [16]. Continuous cropping of some species can also result in declining yields over time. This has been observed in *Panax ginseng* C.A.Mey., whose yields decline due to the proliferation of soilborne disease attributed to the accumulation of root exudates [17]. Hence, reliable methods to improve the secondary metabolite content in cultivated plant material are required.

One potential solution to this is the hydroponic cultivation of medicinal plants. Hydroponics utilises liquid media as the source of micro- and macronutrients that plants require for growth rather than the soil used in traditional systems [18]. There are a number of different systems and techniques used that fall into the category of "hydroponics". Hydroponic techniques have been divided into two categories: solution culture and media culture, and their differences are outlined as follows.

In solution culture, the roots of plants are placed directly into liquid. Solution culture can be further subdivided into circulating and non-circulating systems. In circulating systems, nutrient solution is pumped from a reservoir into a tank, which then flows back into the reservoir. The nutrient film technique (NFT) utilises a tank on a slope which allows a shallow flow of water over the roots of the plant [19]. Contrastingly, in the deep flow technique (DFT), the tray is filled with solution and the roots are completely submerged [20]. Non-circulating systems are not pumped from a reservoir, and the nutrient solution simply sits within the culture tank and is then replaced when the nutrient concentration is inadequate or the pH or electrical conductivity (EC) are unsuitable. Examples of non-circulating systems include the root dipping technique (RDT), floating technique (FT) and capillary action technique. The root dipping technique uses plants suspended over nutrient solution while only the bottom portion of the roots are within the nutrient solution, while the floating technique sees the roots fully submerged [21], analogous to NFT and DFT, respectively. When the capillary action technique is used, nutrient solution is provided either via placing the pot into a very shallow container, where the solution then makes its way through the media by capillary action, or is transported from a reservoir to the media via wicks [22]. Finally, the ebb and flow technique is similar to DFT, but the nutrient solution is periodically drained from the culture tray and then re-added [23].

Solid media culture differs from solution culture in that plants are fully supported by a solid substrate to which nutrient solution is then applied either by surface or sub-surface fertigation. The hanging bag technique involves the hanging of around one-metre-long, media-filled polythene bags above a trough or channel. Plants in net pots are placed into holes cut into the sides of the bags. Nutrient solution is pumped from a reservoir into the top of the bags, where it then makes its way through the media, drips into the trough, and is then returned to the reservoir [24]. When the grow bag technique is used, polythene bags containing media are placed on the ground. Small holes are cut into the bag, and seedlings or seeds are then placed inside. Nutrient solution is then fed along pipes to each plant [19]. The pot technique is similar to the grow bag technique, except that plants are placed within individual pots filled with media and individually fertigated [25]. Lastly, in the trench/trough technique, trenches dug into the ground, or troughs created above ground, are lined with a waterproof material and filled with inert media. Plants are then placed in the trough at intervals and are fed nutrient solution via drip irrigation [20]. Sometimes a drainage pipe is placed at the bottom to allow excess solution to drain from the soil.

When compared with soil cultivation, there is contradictory evidence as to whether hydroponic cultivation increases the growth and secondary metabolite concentration of medicinal plants. For example, Yoshimatsu [26] observed an increase in both biomass accumulation and glycyrrhizin content of *Glycyrrhiza uralensis* Fisch. in hydroponics, Surendran et al. [27] recorded that the *Mentha spicata* L. yield, organic acid concentration, and antioxidant content were higher in hydroponically grown plants, and Duan et al. [28] found that *Trichosanthes kirilowii* Maxim. had greater plant height and number of leaves per plant in hydroponics, as well as a 100% seedling survival rate compared with an 87% survival rate in soil. Conversely, Afreen et al. [29] found that hydroponic cultivation resulted in a reduction of root and shoot growth of *G. uralensis*, while Souret and Weathers [30] observed a lower increase in fresh weight in *Crocus sativus* L. when it was grown in hydroponics. Similarly, Maggini et al. [31] found that the echinacoside content of hydroponically grown *Echinacea angustifolia* DC. was below the 1% standard required by the European Pharmacopeia, and Strzemski et al. [32] remarked that cultivation of *Carlina acaulis* L. in hydroponics resulted in lower carlina oxide content than when it was grown in soil. When maximising the yield and medicinal content of hydroponically grown plants is considered, the selection of hydroponic system is an important factor. Hayden [33] compared the cultivation of various medicinal plants in three different systems: NFT, ebb and flow, and aeroponics (where nutrients are provided from an atomised nutrient solution). It was found that the plants responded differently to each system, allocating resources to different parts of the plant, but that no system was specifically suited to cultivating rhizomes. Facilitating the cultivation of different plant organs is important as secondary metabolites are often concentrated within specific tissues of the plant [34]. It is therefore important to select a system suited to cultivating the specific part of the plant that is desired. This is exemplified by Afreen et al. [29], who speculated that *G. uralensis* grew poorly in a DFT system due to leaching of secondary metabolites into the nutrient solution. Oppositely, Yoshimatsu [26] had great success in cultivating the same plant in an NFT system, producing rhizomes with a high content of medicinal compounds. As a consequence, when cultivating medicinal plants, the hydroponic system should be carefully considered in relation to the part of the plant that is desired.

Publications on the topic of hydroponics are widespread. In a review of hydroponic research trends, Erere et al. [35] recorded a total of 2013 scholarly publications on the topic of hydroponics between 2008 and 2018. However, less than 5% (99) of publications were related to the topic of medicine. In addition, of the 207 publications not written in English, 131 (6.5% of the total studies) were written in Chinese, which makes them difficult to access for a large number of researchers. Examination of these studies may reveal insights into hydroponic medicinal plant cultivation that are seldom explored in the wider literature.

The aim of this article is to examine and synthesise previous research on the hydroponic cultivation of medicinal plants. While there are previous reviews exploring soilless culture of medicinal species, an examination of the literature through the lens of the hydroponic systems and techniques is absent. This review is divided into sections by plant organ, outlining the major methods used in their cultivation and the outcomes of research on the topic, with a focus on traditional medicine. Through this, an overview of how medicinal plants are cultivated is established, identifying opportunities for future cultivation efforts. In a closing section, this article also discusses trends in medicinal hydroponic research, highlighting novel approaches that may warrant further investigation.

## 2. Herbs

Leafy green vegetables are among the most commonly cultivated plants in hydroponic systems. Lettuce (*Lactuca sativa* L.) has been cultivated using DFT [36], NFT [37], and pot technique [37], while spinach (*Spinacia oleracea* L.) has been grown in a floating system [38] but has been said to grow more effectively in solid-substrate systems than liquid culture [39]. Additionally, NFT [40] and capillary technique [41] have been used to cultivate pak choi (*Brassica rapa* subsp. *chinensis* (L.) Hanelt). Production of these leafy vegetables is conducted on a commercial scale in a number of different countries [42], and the techniques used may be applicable to medicinal herbs. Indeed, there is an overlap between herbal plants and food plants, with many of them being used as condiments [20] or as "functional foods" [43]. Most medicinal plants that are cultivated hydroponically are those where the aerial parts of the plants are used, typically the leaves and stems [44]. They are used in a variety of ways: they can be brewed into tea, applied topically, dried and powdered, and eaten fresh or cooked in food. Additionally, pharmaceutical companies may obtain active compounds by extracting and concentrating secondary metabolites directly from the raw plant material [45]. Examples of studies hydroponically cultivating plants where the leaves and stems of plants are used medicinally are shown in Table 1.

**Table 1.** Examples of publications discussing hydroponic cultivation of medicinal plants whose stems or leaves are utilised.

| Hydroponic System | Plant Species | References |
|---|---|---|
| Liquid Culture Methods | | |
| Deep Flow Technique | *Agastache rugosa* Kuntze | [46–50] |
| | *Cannabis sativa* L. | [51] |
| | *Datura stramonium* L. | [52] |
| | *Euphorbia peplus* L. | [53] |
| | *Mentha spicata* L. | [27,54] |
| | *Mentha arvensis* var. *piperascens* Malinv. ex Holmes (synonym of *Mentha canadensis* L.) | [54] |
| Nutrient Film Technique | *Atropa belladonna* L. | [26] |
| | *Cannabis sativa* L. | [51] |
| | *Plectranthus amboinicus* (Lour.) Spreng. | [55] |
| | *Datura stramonium* L. | [52] |
| | *Lepidium sativum* L. | [56] |
| | *Mentha* × *piperita* L. | [33] |
| | *Morus alba* L. 'Ichinose' | [57] |
| | *Nepeta cataria* L. | [33] |
| | *Origanum dictamnus* L. | [58,59] |
| | *Ocimum basilicum* L. | [56] |
| | *Scutellaria lateriflora* L. | [33] |
| | *Urtica dioica* L. | [33] |

**Table 1.** *Cont.*

| Hydroponic System | Plant Species | References |
|---|---|---|
| Floating Technique | *Artemisia vulgaris* L. | [60,61] |
| | *Camellia sinensis* (L.) Kuntze 'Yabukita' | [62,63] |
| | *Cannabis sativa* L. | [51] |
| | *Cannabis sativa* L. 'Cherry', 'Cherry Blossom', and 'Canda' | [64] |
| | *Cannabis sativa* L. 'Pennywise' | [65] |
| | *Cannabis sativa* L. type-II chemovar 'Nordle' and type-I chemovar 'Sensi Star' | [66] |
| | *Camptotheca acuminata* Decne. | [67] |
| | *Centella asiatica* (L.) Urb. | [68] |
| | *Plectranthus amboinicus* (Lour.) Spreng. | [69] |
| | *Coriandrum sativum* L. | [70] |
| | *Dendrobium nobile* Lindl. | [71] |
| | *Ephedra sinica* Stapf | [72] |
| | *Hyssopus officinalis* L. 'Lekar' | [73] |
| | *Ilex purpurea* Hassk. (synonym of *Ilex chinensis* Sims) | [74] |
| | *Leonurus japonicus* Houtt. | [75] |
| | *Lobelia chinensis* Lour. | [76] |
| | *Melissa officinalis* L. | [77] |
| | *Mentha* × *piperita* L. | [78] |
| | *Ocimum basilicum* L. | [31] |
| | *Ocimum basilicum* L. 'Genovese' | [78,79] |
| | *Ocimum basilicum* L. 'Eleonora', 'Aroma 2', and 'Italiano Classico' | [80] |
| | *Ocimum basilicum* L. 'Superbo' and 'Dark Opal' | [79] |
| | *Platycladus orientalis* (L.) Franco | [81–83] |
| | *Solanum nigrum* Acerbi ex Dunal | [76] |
| | *Stellaria media* (L.) Vill. | [60,61] |
| | *Urtica dioica* L. | [84,85] |
| Capillary Action Technique | *Aloe vera* (L.) Burm.f. | [86] |
| | *Cannabis sativa* L. | [51] |
| Ebb and Flow Technique | *Cannabis sativa* L. | [51] |
| | *Mentha* × *piperita* L. | [33] |
| | *Moringa oleifera* Lam. 'PK1' and Malawi' | [87] |
| | *Nepeta cataria* L. | [33] |
| | *Ocimum basilicum* L. 'Emily' | [88] |
| | *Scutellaria lateriflora* L. | [33] |
| | *Urtica dioica* L. | [33] |
| Solid Culture Methods | | |
| Grow Bag Technique | *Cannabis sativa* L. | [51] |
| | *Mentha* × *piperita* L. | [89] |
| | *Moringa oleifera* Lam. 'PK1' and Malawi' | [90,91] |
| Pot Technique | *Aloysia citrodora* Paláu 'Verbena' | [92] |
| | *Cannabis sativa* L. | [51,93] |
| | *Cannabis sativa* L. 'Bialobrzeskie' and 'Monoica' | [94] |
| | *Cannabis sativa* L. 'McLove' | [95] |
| | *Cannabis sativa* L. type-II chemovar 'Nordle' | [96] |
| | *Datura stramonium* L. | [97] |
| | *Helichrysum odoratissimum* Sweet | [98] |
| | *Mentha arvensis* L. | [99] |
| | *Mentha* × *piperita* L. | [99] |
| | *Mentha spicata* L. | [99] |
| | *Ocimum basilicum* L. 'Chádek červená', 'Litra', and 'Mánes' | [99] |
| Trough Technique | *Cannabis sativa* L. | [51] |
| | *Datura stramonium* L. | [100] |

Similar to the cultivation of leafy vegetables, the most commonly used technique among studies growing medicinal plants is FT. This system is effective for the cultivation of plants that have shorter production times, as this system is prone to disease [101]. *Mentha* species [27,99], *Ocimum basilicum* L. [31,88], *Agastache rugosa* Kuntze [46–50], and *Urtica dioica* L. [84] only require around one month from transplant to harvest, while *Artemesia vulgaris* L. and *Stellaria media* (L.) Vill. can be harvested after three months [60]. As there is no reservoir in FT, when the nutrient solution is depleted the culture container must be directly replenished. This is required on around a weekly basis [60,63,70]. The popularity of FT is likely due to the studies focusing on the effects of controlled environments on plant physiology, rather than developing techniques for commercial cultivation. Many of the studies aimed to test the response of the plants to different environmental conditions such as salt stress [66,72,74,81], plant elicitors [64,72,82], endophytic fungi [71], heavy metal exposure [76], nitrogen enrichment [67], and macroelement omission [77]. Responses to these factors may be observed after only a few weeks of exposure, so cultivating plants to maturity may not be required. This leaves room for studies to examine the hydroponic culture of plants that take longer to grow. This is particularly important for the tree species studied—*Camptotheca acuminata* Decne. [67], *Ilex purpurea* Hassk. (now called *Ilex chinensis* Sims) [74], and *Platycladus orientalis* (L.) Franco [81–83]—which may be better cultivated in other hydroponic systems.

While liquid culture appears to be more widely used for small leafy plants, medicinal trees and shrubs used for their leaves are hydroponically cultured using solid media systems. This is because taller plants require a solid substrate to support their weight [21]. Additionally, the basal area of the stem needs to be kept dry to prevent rot [87]. Young *Moringa oleifera* Lam. trees cultivated in grow bags filled with sand and coir at a 3:1 ratio and fertigated with nutrient solution had greater secondary metabolite content when compared with those grown in the field [91]. Likewise, the fresh weight and leaf number of *Aloysia citrodora* Paláu grown in pots containing a 3:1 mixture of perlite and sand supplied with nutrient solution were also greater than those of their soil-grown counterparts [92]. The selection of sand as a significant component of the culture medium for both experiments suggests that adequate drainage is important for larger plants. These systems may be extended for use in cultivating other medicinal trees. In particular, the grow bag system used for the cultivation of *M. oleifera* [90,91] shows promise due to its simple design.

Studies focused on commercial cultivation use a variety of techniques. One particular focus is on the use of plant factories with artificial lighting (PFALs). The aim of PFALs is to optimise environmental conditions to maximise growth and phytonutrient content of high-value plants like vegetables and medicinal plants [102]. A series of studies on the cultivation of *A. rugosa* were conducted with the intention of optimising environmental conditions for commercial production in a PFAL [46–50] using DFT. Similarly, Bafort et al. [53] also used DFT in determining optimum growth conditions and assessing the economic viability of producing *Euphorbia peplus* L. in a modified shipping container analogous to a PFAL. Two different hydroponic techniques—NFT and ebb and flow—were compared by Hayden et al. [33] for the purposes of developing commercial cultivation. They found that *Mentha × piperita* L. and *Nepeta cataria* L. grew well in the ebb and flow system, while both systems were less effective than aeroponics for a variety of species. The use of FT has also been proposed for the commercial production of *O. basilicum* [31], as well as *A. vulgaris and S. media* [60], as it has low start-up costs and is cheap to run [45].

*Cannabis sativa* L. has been extensively studied due to its popularity as a recreational and medicinal drug. In addition, its high economic value means that cannabis cultivation is at the forefront of innovation with regards to the adoption of modern commercial production techniques for plants [103]. It has been cultivated in numerous hydroponic systems, but ebb and flow and NFT are among the most widely used and high-yielding systems [51]. These factors make studies on *C. sativa* cultivation valuable resources for determining the best practices for growing leafy medicinal plants.

### 3. Flowers

　　Many kinds of flowers are sought-after for their aesthetic, nutritional, or medicinal properties. Of these categories, ornamental flowers such as roses, gerberas, chrysanthemums, carnations, and lilies are those most commonly grown on a commercial level in hydroponic systems [25]. Commercial production of ornamental flowers uses a wide selection of different hydroponic systems such as DFT [104], NFT [105–108], trough [107,109], and grow bag [101,109]. However, cut and potted flowers are usually grown in solid culture substrates [20,101]. Flowers used for nutritional purposes have also been cultivated hydroponically. However, in contrast to ornamental plants, liquid culture appears to be preferred, with DFT [110–112] and NFT [112,113] being popular techniques. Flowers are the least frequently used plant organ for medicinal purposes [114]. Despite this, there are still many flowers used medicinally in the form of essential oils, juices, and teas, as well as whole in their fresh or dried forms [115]. Species often fall into more than one category. *Chrysanthemum × morifolium* (Ramat.) Hemsl. is used as an ornamental plant but is also prominent in TCM, while *Calendula officinalis* L. can be used to treat skin disorders or eaten in salads [116]. Therefore, there may be similarities in the hydroponic cultivation methods of plants across these three categories. Table 2 shows examples of studies where medicinal flowers were hydroponically cultivated.

**Table 2.** Examples of publications discussing hydroponic cultivation of medicinal plants whose flowers are utilised.

| Hydroponic System | Plant Species | References |
|---|---|---|
| Liquid Culture Methods | | |
| Deep Flow Technique | *Chrysanthemum × morifolium* (Ramat.) Hemsl. 'Syuho-no-chikara' | [104] |
| | *Taraxacum officinale* F.H.Wigg. | [117] |
| Nutrient Film Technique | *Crocus sativus* L. | [30] |
| | *Datura stramonium* L. | [52] |
| | *Hypericum perforatum* L. 'Topas' | [118] |
| | *Hypericum perforatum* L. 'New Stem' | [119] |
| | *Taraxacum officinale* F.H.Wigg. | [117] |
| Floating Technique | *Achillea millefolium* L. | [60,61,120–122] |
| | *Borago officinalis* L. | [61,122] |
| | *Calendula officinalis* L. | [60,61,123] |
| | *Carthamus tinctorius* L. | [124,125] |
| | *Carthamus tinctorius* L. 'Zarghan-Fars' | [126] |
| | *Carthamus tinctorius* L. 'PBNS-12' | [127] |
| | *Crocus sativus* L. | [128,129] |
| | *Chrysanthemum × morifolium* (Ramat.) Hemsl. 'Wuyuanhuang' | [130] |
| | *Datura innoxia* Mill. | [131–133] |
| | *Datura stramonium* L. | [52] |
| | *Hypericum olympicum* L. | [134] |
| | *Hypericum orientale* L. | [134] |
| | *Hypericum perforatum* L. | [61,134–136] |
| | *Lonicera japonica* Thunb. | [137,138] |
| | *Prunella vulgaris* L. | [139] |
| | *Sophora japonica* L. (synonymous with *Styphnolobium japonicum* (L.) Schott) | [74,140,141] |
| | *Tanacetum parthenium* Sch.Bip. | [60,61,121] |
| | *Taraxacum officinale* F.H.Wigg. | [60,61,121] |
| Capillary Action Technique | *Chrysanthemum × morifolium* (Ramat.) Hemsl. 'Syuho-no-chikara' | [104] |
| | *Pelargonium graveolens* L'Hér. | [142] |
| Ebb and Flow Technique | *Crocus sativus* L. | [128,143] |
| | *Hypericum perforatum* L. 'New Stem' | [144] |

**Table 2.** *Cont.*

| Hydroponic System | Plant Species | References |
|---|---|---|
| Solid Culture Methods | | |
| Pot Technique | *Calendula officinalis* L. | [123] |
| | *Crocus sativus* L. | [145,146] |
| | *Datura innoxia* Mill. | [97] |
| | *Datura metel* L. | [97] |
| | *Datura sanguinea* (now known as *Brugmansia sanguinea* (Ruiz & Pav.) D.Don) | [97] |
| | *Datura stramonium* L. | [97] |
| | *Hypericum perforatum* L. | [147] |
| | *Hypericum perforatum* L. 'Topas' | [148] |
| | *Lavandula angustifolia* Mill. | [147] |
| | *Lonicera japonica* Thunb. | [149] |
| | *Sophora japonica* (synonymous with Styphnolobium japonicum (L.) Schott) | [141] |
| | *Thymus vulgaris* L. | [147] |
| Trough Technique | *Crocus sativus* L. | [150] |
| | *Datura stramonium* L. | [100] |

As with herbal plants, FT is one of the most common systems used to hydroponically grow flowers. Similar to the studies on leafy medicinal plants, studies cultivating medicinal flowers used FT to examine the response of plants to controlled environmental conditions, particularly salinity [74,124,125,137,138,141], plant growth-promoting rhizobacteria [52,132,133], heavy metal contamination [126,127,134], drought stress [137,140], and plant elicitors [136,138]. Hydroponic cultivation was usually carried out for less than two months in the above studies, further reinforcing the idea that FT is primarily for experimental growth rather than commercial production. Regardless, there are examples of commercial flower production using FT. Dorais et al. [60] raised *Achillea millefolium* L., *Borago officinalis* L., *C. officinalis*, *Tanacetum parthenium* Sch.Bip., and *Taraxacum officinale* F.H.Wigg. in an FT system and was able to harvest marketable raw material within four months, while Ai et al. [130] produced *C. × morifolium* in 160 days.

Capillary technique is said to be effective for the cultivation of flowers [24], yet studies utilising the technique are rare. The differences in the characteristics that are desired in cut or ornamental flowers and those used for medicinal purposes may be the cause of this. Chrysanthemum plants grown in a capillary mat system were taller, had stronger stems, and a longer vase life than those grown in a DFT system [104]. In flowers grown for aesthetic purposes, these are important factors in determining the quality of a flower. Yet in the same study, a DFT system with intermittent cycling of nutrient solution resulted in the greatest fresh weight in flowers, which is more important when producing material for medicinal use. Another study comparing hydroponic methods also supported the effectiveness of DFT for cultivating species utilized for their flowers. In the cultivation of dandelion (*T. officinale*) plants using either NFT or DFT, it was found that root and shoot yield was greater when DFT was used, although it is worth noting that plants were not cultivated until flowering occurred [117]. Studies examining other liquid culture methods showed mixed results. Saffron (*C. sativus*) plants grown using NFT showed reduced incidence of flowering and reduced biomass compared with those grown in aeroponics or soil, while the stigmata dry weight—the medicinally and nutritionally valuable part of the flower—was greater [30]. Conversely, *H. perforatum* flowers harvested from an NFT system had hypericin, hyperforin, and pseudohypericin concentrations greater than or equivalent to field-grown plants [119]. Ebb and flow systems were also effective, resulting in increased phytochemical content in *H. perforatum* [144], and controlled-environment cultivation of saffron with a similar system has also been shown to be effective [128,143].

Among studies using solid culture techniques to grow flowers, only two techniques are prominent—trough and pot. Substrate culture helps provide structural support to the plants, facilitating the growth of taller plant species [24]. Studies examining trough

technique have contradictory findings. A comparison between trough technique and soil culture showed that *Datura stramonium* L. plants responded better to the hydroponic system, resulting in taller plants with greater root development [100], while saffron had a greater yield under soil cultivation [150]. However, the initial observations of Sellar [100] were that soil-grown and soilless *D. stramonium* plants were initially comparable in growth and that the hydroponically cultured plants showed greater yield upon maturation. Conversely, other studies have shown that initial corm size is a determining factor on the yield of saffron [128,151,152], which may suggest that the yield of some flower species may be determined by their state prior to transplantation into hydroponics. Pot technique is the solid culture method most commonly used in publications. In contrast to when many smaller herbal parts are used, the flowers may be harvested over many growing seasons and may have different requirements at different growth stages. The use of pots allows plants to be moved as they grow, enabling them to be cultivated in close proximity during the seedling stage and then spaced apart as they mature, using the cultivation area more efficiently while reducing competition and improving growth [153]. Plants grown in pots can also be transferred to larger ones as they grow; however, the amount of solution required to sustain the plant may change, and care should be taken to ensure that the substrate holds adequate water and that it is not draining from the container [51].

## 4. Roots and Rhizomes

Interest in producing food for manned space missions has resulted in research investigating the hydroponic cultivation of plants where underground parts are used, such as potatoes, sweet potatoes, and peanuts [154]. These studies have used both liquid and solid culture methods, including NFT [155,156], capillary action technique [157,158], and trough technique [159]. Despite this, it has been observed that hydroponic systems used to culture above-ground plant organs are generally unsuited to cultivating roots, rhizomes, and stolons [26]. These parts—rhizomes and stolons in particular—fail to properly develop when plants are fully submerged in water [158]. Consequently, systems that allow the lower part of the stem and upper part of the root to be above the water level have been developed. Air-gap methods have been used effectively in both substrate culture [157] and liquid culture [160]. However, solid culture methods may not be efficient for the commercial cultivation of underground parts. Growth of tubers in rockwool breaks up the substrate due to its fragility [154], while roots may penetrate expanded clay pellets—reducing harvestable biomass and possibly harbouring sources of infection [161]. Solid culture substrates therefore have to be replaced between cultivation cycles, increasing production costs.

Examples of the hydroponic cultivation of roots or rhizomes can be seen in Table 3. For cultivation of below-ground plant organs, pot technique is a common solid culture method. Various media are used throughout the literature, including perlite [162,163], peat moss [162], sand [164], rockwool [165], quartz sand [166], and clay pebbles [167]. Ahmadi et al. [162] compared the growth of *Echinacea purpurea* (L.) Moench in different ratios of pearlite and peat moss and different $NO_3^-$:$NH_4^+$ ratios. A 1:1 ratio of perlite to peat moss at a 90:10 nitrate to ammonia ratio resulted in the highest fresh root weight per plant. The authors proposed that this mixture results in a medium with a high water-holding capacity, while also maintaining a high air-filled porosity—supplying plants with both water and nutrients while also aerating the roots of the plant. Comparisons between solid and liquid culture techniques showed differing outcomes. Cultivation of *Echinacea angustifolia* DC. had higher cichoric acid content and *E. purpurea* achieved the greatest secondary metabolite concentrations when grown in DFT compared with sand and Pro-Mix [164]. Furthermore, *Valeriana officinalis* L. grown using FT had greater root dry weight than when grown using a 1:1 mixture of perlite and vermiculite [168]. Conversely, growth of *M. plumbea* had increased root and shoot growth when grown in pots containing perlite than when grown using FT [163]. This may be due to fine roots and other below-ground organs responding differently to being submerged in nutrient solution. Therefore, studies using liquid culture to grow underground parts are also important to consider.

**Table 3.** Examples of publications discussing hydroponic cultivation of medicinal plants whose roots or rhizomes are utilised.

| Hydroponic System | Plant Species | References |
|---|---|---|
| Liquid Culture Methods | | |
| Deep Flow Technique | *Astragalus membranaceus var. mongholicus* (Bunge) P.K.Hsiao (synonym of *Astragalus mongholicus* Bunge) | [169] |
| | *Echinacea purpurea* (L.) Moench | [164] |
| | *Echinacea angustifolia* DC. | [164] |
| | *Glycyrrhiza glabra* L. | [170] |
| | *Glycyrrhiza uralensis* Fisch. | [29] |
| | *Ligularia fischeri* Turcz. | [171] |
| | *Picrorhiza kurroa* Royle ex Benth. | [172] |
| | *Taraxacum officinale* F.H.Wigg. | [117] |
| Nutrient Film Technique | *Anemopsis californica* (Nutt.) Hook. & Arn. | [33] |
| | *Arctium lappa* L. | [33] |
| | *Atropa belladonna* L. | [26] |
| | *Coptis japonica* Makino | [26] |
| | *Morus alba* L. 'Ichinose' | [57] |
| | *Glycyrrhiza uralensis* Fisch. | [26] |
| | *Glycyrrhiza uralensis* Fisch. 'GuIV1' | [173,174] |
| | *Glycyrrhiza uralensis* Fisch. 'GuIV1' and 'GuIV2' | [175] |
| | *Taraxacum officinale* F.H.Wigg. | [117] |
| | *Urtica dioica* L. | [33] |
| | *Withania somnifera* (L.) Dunal | [176] |
| | *Zingiber officinale* Roscoe | [33] |
| Floating Technique | *Astragalus membranaceus* Fisch. ex Bunge (synonymous with *Astragalus mongholicus* Bunge) | [177,178] |
| | *Carlina acaulis* L. | [32] |
| | *Coptis chinensis* Franch. | [179,180] |
| | *Echinacea angustifolia* DC. | [31,181] |
| | *Glycyrrhiza uralensis* Fisch. | [182,183] |
| | *Inula helenium* L. | [60,61,121] |
| | *Ligularia fischeri* Turcz. | [171] |
| | *Lycium chinense* Mill. | [184,185] |
| | *Merwilla plumbea* (Lindl.) Speta | [163] |
| | *Panax notoginseng* (Burkill) F.H.Chen ex C.Y.Wu & K.M.Feng | [166,186] |
| | *Pueraria montana* (Lour.) Merr. | [187] |
| | *Pueraria montana var. lobata* (Willd.) Maesen & S.M.Almeida ex Sanjappa & Predeep | [188] |
| | *Rehmannia glutinosa* (Gaertn.) DC. | [189,190] |
| | *Salvia miltiorrhiza* Bunge | [191] |
| | *Taraxacum officinale* F.H.Wigg. | [60,61,121] |
| | *Trichosanthes kirilowii* Maxim. | [28] |
| | *Urtica dioica* L. | [84,85] |
| | *Valeriana officinalis* L. | [60,61,168] |
| Ebb and Flow Technique | *Anemopsis californica* (Nutt.) Hook. & Arn. | [33] |
| | *Arctium lappa* Willd. | [33] |
| | *Bupleurum falcatum* L. | [192] |
| | *Urtica dioica* L. | [33] |
| | *Zingiber officinale* Roscoe | [33] |
| Solid Culture Methods | | |
| Pot Technique | *Echinacea purpurea* (L.) Moench | [162,164] |
| | *Echinacea angustifolia* DC. | [164] |
| | *Glycyrrhiza glabra* L. | [165] |
| | *Merwilla plumbea* (Lindl.) Speta | [163] |
| | *Panax notoginseng* (Burkill) F.H.Chen ex C.Y.Wu & K.M.Feng | [166] |
| | *Scutellaria baicalensis* Georgi | [167] |
| | *Valeriana officinalis* L. | [168] |

Liquid culture techniques are more widely varied, and include DFT, NFT, FT, and ebb and flow technique. In general, species where roots were utilised medicinally grew well in the liquid culture systems when compared with other cultivation methods. *Astragalus membranaceus* var. *mongholicus* (Bunge) P.K.Hsiao (now known as *Astragalus mongholicus* Bunge) cultivated for four weeks using DFT had greater astragaloside IV content than those grown for two years in soil [169], and *Withania somnifera* (L.) Dunal plants grown using NFT had an average dry weight and withaferin A concentration larger than those grown using aeroponics [176]. Similarly, the use of FT to cultivate *Inula helenium* L. and *T. officinale* resulted in increased root dry weight when compared with those grown in the field [60]. *Ligularia fischeri* Turcz. grown using DFT had increased root fresh weight, as well as antioxidant capacity, compared to their soil-grown counterparts [171]. Additionally, the FT culture of *E. angustifolia* was estimated to be able to produce a root dry weight equal to 3–4 years of field production in 32 weeks [31]. In a comparison of ebb and flow technique and soil cultivation of *Bupleurum falcatum* L., no significant differences were found between the hydroponic and soil-grown plants with regards to both chemical content and yield [192]. When examining the differences between liquid culture methods, a comparison of *T. officinale* plants grown in either DFT or NFT found that DFT resulted in the highest root dry weight as well as root:shoot ratio [117]. However, there were exceptions to this. While hydroponic *I. helenium* and *T. officinale* had increased biomass, their secondary metabolite contents were decreased. The phenolic content of *T. officinale* was 6.2 times larger when grown in soil, and the sesquiterpene lactone content of field-grown *I. helenium* was more than double that of the hydroponically cultivated samples [121]. Hydroponic *C. acaulis* was also found to have lower carlina oxide and antioxidant activity when compared with those grown in soil [32]. This suggests that while hydroponic cultivation often results in increased root biomass, species respond differently to hydroponics in terms of secondary metabolite content, and so the efficiency of hydroponic production should be judged on a species-by-species basis.

In contrast, the cultivation of rhizomes and bulbs in liquid culture is generally less successful. In a study comparing different hydroponics systems, Hayden [33] found that neither NFT nor ebb and flow technique were ideal for rhizome production and that in NFT the production of roots was drastically reduced. The use of DFT also resulted in low root dry weight when cultivating *Glycyrrhiza glabra* L. [170] and *G. uralensis* [29]. Likewise, *M. plumbea* cultivated using FT grew poorly and a minimal increase in root and bulb weight were observed [163]. Some plants whose rhizomes are used medicinally have been shown to grow well in liquid culture. Valerian (*V. officinalis*) cultivated in a floating system has been shown to produce a greater root dry mass than when cultivated in aeroponics and mixed perlite–vermiculite hydroponics [168], as well as soil [60,61], while *Picrorhiza kurroa Royle ex Benth.* plantlets—grown both from seed and from tissue culture—showed good rootlet growth [172]. However, despite these results, the plants shown in the figures of both Tabatabaei [168] and Thakur et al. [172] show reduced rhizome development, and Dorais et al. [60] makes no mention of the presence of rhizomes, suggesting that while root growth may be improved, growth of the medicinally valuable rhizome is reduced. Comparably, studies have shown FT to be an effective system for the growth of *Coptis chinensis* Franch. [179,180], *G. uralensis* [182], and *Panax notoginseng* (Burkill) F.H.Chen ex C.Y.Wu & K.M.Feng [166,186]; however, these studies utilised plants taken from the wild with rhizomes already present from growth in soil, making it difficult to compare the results with those of other studies where plants were cultivated hydroponically from seedlings.

Overall, research suggests that below-ground storage organs struggle to grow when immersed in nutrient solution, while fibrous roots are able to survive as long as nutrient solution is sufficiently aerated [157]. Air stones within liquid culture techniques may therefore provide enough dissolved oxygen to maintain roots, but other organs such as rhizomes require a cultivation method that prevents waterlogging [158], and solid culture can provide the necessary air space. In response to this, air-gap techniques have been developed. Hayden [33] argues that of the hydroponic systems tested in their study,

a perlite air-gap system comprised of a reservoir of perlite with a shallow layer of nutrient solution at the bottom—akin to NFT but with solid media—showed the most promise with regards to future work examining rhizome cultivation. The air-gap system has been shown to be effective in cultivating *G. uralensis*, *C. chinensis*, and *Atropa belladonna* L. (the former two having rhizomes, while the latter has fleshy roots) with the resulting plant matter containing secondary metabolite concentrations above the requirements of the Japanese Pharmacopoeia [26]. However, due to the previously mentioned disadvantages of cultivating underground plant organs in substrate culture, there is a desire to create an air-gap system without solid media. Such a system was designed by Sawada et al. [193], using cylindrical tubes to suspend the bottom half of roots in nutrient solution while also providing an air-gap for the growth of rhizomes. This system has been shown to produce *G. uralensis* plants with extracts comparable in both safety and efficacy to the commercial crude drug [173–175]. Similar methods may be useful for the commercial cultivation of rhizomes for production of pharmaceuticals or for use in traditional medicine.

## 5. Fruits and Seeds

Fruit crops are among the most widely hydroponically cultivated plants in the world, including tomatoes, cucumbers, sweet peppers, and melons [25]. Commercial soilless production of fruit utilises a variety of techniques. Tomatoes and cucumbers have been grown commercially using trough and pot techniques with different substrates, while peppers have been grown using the growbag technique [101]. Cucumbers and tomatoes are also reported to grow well in DFT [22], and a comparison of tomatoes grown using FT, pot technique with a rockwool substrate, and soil found that the fruit yields were not significantly different but that lycopene and β-carotene concentrations were greater in fruits grown with FT [194]. NFT is said to provide an optimal environment for the cultivation of strawberries, blueberries, and melons [20], but it has also been suggested that growing strawberries in solid media culture with perlite, vermiculite [195], or coconut fibre [196] is effective. Solid culture methods may provide the support required by larger climbing plants, while lower-growing plants benefit from liquid culture methods, enabling similar methods to be used when cultivating medicinal fruits. Furthermore, there is also an overlap between some fruits used for medicinal purposes and edible fruits. Cranberry (*Vaccinium macrocarpon* Aiton), grape (*Vitis vinifera* L.), and goji (*Lycium barbarum* Mill./ *Lycium chinense* Mill.) are among the top-selling herbal supplements in the Unites States and Europe [197], and jujube (*Ziziphus jujuba* Mill.) is used as both a food and in TCM [198]. Examining techniques used to cultivate fruits consumed as food may therefore also aid with designing systems for cultivating medicinal fruit.

Among studies examining the hydroponic cultivation of medicinal fruits, the most commonly used technique is FT (Table 4). As with the other plant organs examined, studies growing plants whose fruits or seeds are used medicinally with FT focused on short-term cultivation to test the response of plants to different environmental conditions. The most commonly studied environmental factors were salinity [66,74,81,184,185,199,200] and plant elicitors [64,82,199,201]. However, in these studies the plants were not cultivated until they produced fruit, and FT is therefore difficult to recommend for commercial production.

**Table 4.** Examples of publications discussing hydroponic cultivation of medicinal plants whose fruits or seeds are utilised.

| Hydroponic System | Plant Species | References |
|---|---|---|
| Liquid Culture Methods | | |
| Deep Flow Technique | *Cannabis sativa* L. | [51] |
| Nutrient Film Technique | *Cannabis sativa* L. <br> *Silybum marianum* (L.) Gaertn. | [51] <br> [202] |

**Table 4.** *Cont.*

| Hydroponic System | Plant Species | References |
|---|---|---|
| Floating Technique | *Ammi visnaga* (L.) Lam. (synonym of *Visnaga daucoides* Gaertn.) | [200] |
| | *Cannabis Sativa* | [51] |
| | *Cannabis sativa* L. 'Cherry', 'Cherry Blossom', and 'Canda' | [64] |
| | *Cannabis sativa* L. 'Pennywise' | [65] |
| | *Cannabis sativa* L. type-II chemovar 'Nordle' and type-I chemovar 'Sensi Star' | [66] |
| | *Gardenia jasminoides* J.Ellis | [203] |
| | *Ligustrum lucidum* W.T.Aiton | [74] |
| | *Lycium chinense* Mill. | [184,185] |
| | *Lycium barbarum* Mill. | [184,185] |
| | *Morus alba* L. 'Ichinose' | [57] |
| | *Passiflora suberosa* L. | [201] |
| | *Platycladus orientalis* (L.) Franco | [81–83] |
| | *Trichosanthes kirilowii* Maxim. | [28] |
| | *Vaccinium macrocarpon* Aiton 'Stevens' | [204] |
| | *Ziziphus jujuba* Mill. | [74,199] |
| Capillary Action Technique | *Cannabis sativa* L. | [51] |
| Ebb and Flow Technique | *Cannabis sativa* L. | [51] |
| Solid Culture Methods | | |
| Grow Bag Technique | *Cannabis sativa* L. | [51] |
| Pot Technique | *Cannabis sativa* L. | [51,93] |
| | *Cannabis sativa* L. 'Bialobrzeskie' and 'Monoica' | [94] |
| | *Cannabis sativa* L. 'McLove' | [95] |
| | *Cannabis sativa* L. type-II chemovar 'Nordle' | [96] |
| | *Silybum marianum* (L.) Gaertn. | [205] |
| | *Vitis vinifera* L. 'Malbec' and 'Négrette' | [206,207] |
| | *Vitis vinifera* L. 'Fer Servadou', 'Tannat', and 'Duras' | [206] |
| Trough Technique | *Cannabis sativa* L. | [51] |
| | *Gardenia jasminoides* J.Ellis 'Heaven Scent' | [208] |

Another commonly utilised technique is the pot technique. In these studies, a variety of different media were used, including rock wool [93], perlite [94], clay balls [93,95], peat [96], and pozzuolana (volcanic rock fragments) [206,207]. Combinations of media were also evidenced by [205], who utilised a 1:1 mixture of sand and peat moss. Of these studies, only three cultivated plants to the point where they bore fruit. Hydroponic cultivation of red grape wine cultivars in pozzuolana had differing potassium content in both leaves and fruits [206], while the grafting of different rootstocks resulted in differences in organic acid content [207]. *Silybum marianum* (L.) Gaertn. also bore fruit when cultivated using the pot technique, yielding significantly more fruit when sprayed with $10^{-4}$ M of salicylic acid [205]. NFT was also shown to be effective when growing *S. marianum*, resulting in an increased number of seeds per flower-head compared with field-grown plants [202]. Exposure to 200 μM of salicylic acid further increased the number of seeds. This suggests that the hydroponic systems used in commercial edible fruit production may be applicable to medicinal fruit—in particular with larger plants—as fruit tree seedlings of pears, peaches, and tangerines have also been cultivated in solid culture systems [195].

Of all the plants cultivated hydroponically, cannabis is one of the most frequently examined. As previously discussed, the leaves are most commonly used for medicinal extracts [209]. However, other parts of *C. sativa* are also medicinally valuable—the fruits and seeds are used in TCM [210], while the seeds are also used in the production of hemp seed oil [211]. Furthermore, cannabis cultivars also used for the production of TCM and hemp seed oil differ from those used to medicinal cannabis. Modern medical cannabis uses drug-type cultivars which are bred to maximise content of $\Delta^9$-tetrahydrocannabinolic acid and cannabidiolic acid [96]. In contrast, the fruits and seeds used in TCM are from fibre-type

cultivars [212], as are those from which hemp seed oil is produced [211]. While most of the literature focuses on drug-type cultivars, some examples of hydroponic cultivation of fibre-type cultivars exist. Bailey [64] examined the effect of the plant elicitors methyl jasmonate and salicylic acid on industrial hemp varieties ('Cherry', 'Cherry Blossom', and 'Canda') and found that methyl jasmonate was the most effective at increasing the production of cannabidiolic acid and cannabichromene in leaves, while Kalousek et al. [94] measured the effect of landfill leachate on different varieties of fibre-type hemp ('Bialobrzeskie' and 'Monoica') and observed that the application of leachate decreased the aerial dry weight and total leaf area of the plants. Despite these studies focusing on fibre-type *C. sativa*, neither of them cultivated plants to the stage where they produced fruit, providing opportunity for further research into the environmental factors that affect the medicinal qualities of fruit or seeds.

## 6. Discussion

The investigation of publications demonstrating hydroponic cultivation of different plant organs revealed differences in the effectiveness of different hydroponic techniques. Liquid culture techniques such as DFT and NFT have been shown to be effective at cultivating herbal parts of plants. Flowers grow well when ebb and flow technique is used, as well as with the solid culture methods of the trough and pot techniques. Examples of both solid and liquid culture techniques have been shown to be effective at cultivating roots, while the air-gap NFT technique has been proposed as an efficient way of growing rhizomes. Among studies investigating the growth of fruits, the pot technique is commonly used. However, there were also commonalities among the studies with regards to the cultivation of different plant organs, as well as novel ideas that may be worthy of future study.

Of all the hydroponic techniques, FT is the most frequently used by studies cultivating medicinal plants. While it has been demonstrated as a method for the commercial production of a selection of crops, it is more frequently used to examine the effects of environmental factors on plant growth. Studies cultivating medicinal plants using FT have investigated a number of environmental conditions, including salinity, drought stress, plant hormones, heavy metal contamination, macroelement omission, and interactions with rhizobacteria or endophytic fungi. This is due to the advantages offered by hydroponics when compared with soil culture. The use of FT eliminates the effects of soil-borne pests and diseases, removes the need for irrigation, and increases access to the root zone [213]. Roots can be easily monitored in FT, making it an important technique for studies examining plant physiology. FT is also a relatively straightforward technique as there is no circulation of nutrient solution, making it simple to set up and low-cost in comparison to other techniques [20]. Studies utilising FT tend to be short-term, cultivating plants for only a few weeks up to a period of two months. This is likely due to the risk of disease when cultivating over longer periods. While circulating hydroponic methods can have a variety of methods to sterilise nutrient solution—ultra-violet radiation, filtration, ozone treatment, temperature treatment—it is more difficult to treat the solution of a non-circulating system, and so the outbreak of disease is a greater risk [214]. Consequently, when plants can be grown and harvested quickly, economic loss due to infected plants is reduced, making the use of FT more commercially viable.

Despite this, studies demonstrating long-term cultivation of medicinal plants in FT are also present in the literature. *U. dioica* has been cultivated for a total of 242 days in an unheated greenhouse using an FT system [84]. Over the growing period, the leaves of plants were harvested a total of five times, including during months when the plant is usually dormant when grown in the field. Saito et al. [63] cultivated tea plants using FT for 180 days before harvesting their roots. Cultivation of *C. × morifolium* for 160 days in FT was carried out for flower production by Ai et al. [130]. *V. officinalis* roots grown over 150 days in a floating system had a greater root dry weight and oil content than plants grown in aeroponics, pot technique, or in soil [168]. Similarly, Dorais et al. [60] harvested *V. officinalis* and *I. helenium* roots after 120 days of growth, resulting in yields over four

times that of field-cultivated plants in the same length of time. These studies suggest that while FT has a risk of disease, it is still possible to cultivate plants for longer periods of time—even attaining multiple harvests from the same plants. Consequently, the use of FT should not be restricted to experimental or short-term use and can also be a valuable technique for plant production over longer timescales.

Traditional medicines often use only specific plant organs in their formulae due to the localisation of active compounds in specific parts of a plant. Despite this, studies have also shown that medicinally useful compounds can be found in tissues other than those traditionally harvested. The medicinal value of plants of the *Datura* genus in medicinal treatments has been known since ancient times across a wide variety of cultures [52]. The plants contain the valuable secondary metabolites atropine, hycoscyamine, and scopolamine—members of the tropane class of alkaloids [97]. The flowers of *Datura metel* Mill. are used in TCM [215], while *Datura innoxia* Mill. roots were utilised in medicinal preparations by the native American Chumash [216], and *D. stramonium* leaves have been traditionally used in Europe [217]. In Indian Ayurvedic medicine, the leaves, flowers, and seeds of various *Datura* species are employed for medicinal purposes [218]. However, all parts of the plants contain medicinal compounds, with alkaloids constituting 0.26% of the total weight of the plant [219]. Similarly, *Camellia sinensis* (L.) Kuntze is typically grown for the production of tea from its leaves which, in addition to being a beverage, are used medicinally in TCM [210]. The plant is an almost unique source of the amino acid theanine, save for only one inedible species of mushroom [62]. However, the hydroponic cultivation of *C. sinensis* found that the highest concentration of theanine was found in the lignified taproots, while root tips also contained a significant concentration [63]. St. John's Wort (*H. perforatum*) is a medicinal plant known for the red oil produced from its flowers that is used to treat anxiety and depression in the traditional medicine of a number of European countries [220]. It contains the secondary metabolites hypericin, pseudohypericin, hyperforin, and hyperoside [221]. Despite the flower being the most prominent used part in traditional medicine, the glands that produce the medicinally valuable chemicals are also present on leaves [148], and research has demonstrated that these glands contain comparable concentrations of hypericin and pseudohypericin regardless of the plant organ they are located on [222]. Hydroponic production has been shown to increase the number of glands per leaf, as well as leaf secondary metabolite content [144]. Therefore, while a specific plant organ may be traditionally utilised for treatment, extraction of medically useful chemicals may be possible from many parts of the plant, and hydroponic cultivation may facilitate access to parts of the plant normally difficult to harvest when field cultivated.

The use of liquid media culture offers unique advantages in plant cultivation. Nutrient solution can be applied directly to the roots, and its content can be optimised to enable increased production of biomass or secondary metabolites. Studies examining the effects of varying concentrations of micro- and macronutrients are common in research examining the hydroponic cultivation of plants, as well as other qualities of nutrient solutions such as pH and EC [213]. Although the manipulation of nutrient solution composition is prevalent in the literature, there are also reports on other unique additions to liquid media. Vimolmangkang et al. [54] investigated the effects of the addition of sulphur, amino acids, or a combination of sulphur and amino acids to nutrient solution on the growth and volatile oil content of *M. spicata* and *Mentha arvensis* var. *piperascens* Malinv. ex Holmes (synonym of *Mentha canadensis* L.) using DFT. It was found that *M. spicata* plants yielded the largest weight of volatile oil per gram of fresh weight when provided with nutrient solution supplemented with only the amino acid mixture, while the yield of *M. arvensis* var. *piperascens* plants was greatest when a combined solution of amino acids and sulphur was applied. Additionally, it has been proposed that chemical precursors to desirable secondary metabolites could be added to nutrient solution in order to increase their production. The addition of L-proline and L-ornithine—precursors of stachydrine hydrochloride—to nutrient solution used to cultivate *Leonurus japonicus* Houtt. in FT found that the exposure to 0.5 mmL$^{-1}$ of L-proline for 96 h prior to harvest increased stachydrine hydrochloride

content by more than 50% [75]. Further study into supplementing hydroponic nutrient solutions may provide other methods of increasing secondary metabolite content beyond what optimisation of nutrients can provide. However, the identification of precursors that are cost-effective while also being able to pass from the solution into the roots to be utilised by the plant is an important factor in the viability of the addition of precursors. Alternatively, liquid culture may not require that plants are harvested at all. When plants are cultivated hydroponically, secondary metabolites are often exuded from the roots into the nutrient solution. The extraction of exuded compounds from hydroponic solutions may offer an alternative to the harvesting of plants. Paponov et al. [136] observed that when *H. perforatum* was sprayed with 10 μM of methyl jasmonate, hypericin in root exudates was 88% greater than that of controls. Furthermore, there is evidence that these two techniques can be used in combination. *D. innoxia* grown in a hydroponic solution containing the precursors phenylalanine and ornithine resulted in a significant increase in hyoscyamine and scopolamine present in root exudates when roots were permeabilised with Tween20 [131]. The permeabilisation of roots may facilitate the entry of secondary metabolite precursors into root cells, enabling the synthesis of the metabolites and their subsequent exudation into the liquid media. These metabolites can then be extracted from the media for use in pharmaceutical production.

If the hydroponic cultivation of medicinal plants is to replace wild harvesting or field cultivation it must be commercially viable. In order to determine if production is economical, expenses such as rent, labour, packaging, taxes, electricity, water, and plant materials must be compared with the yield and sales price of the plant. In general, these types of evaluations of hydroponic farms are rare [223], but some examples do exist. Papadopoulos et al. [224] evaluated the cost of an NFT system growing lettuce and tomatoes and a solid media system growing flowering pot plants and tomatoes in a greenhouse in Western Macedonia, Greece. Over a four-year period, they calculated the average operating expenses and gross income of the greenhouse. While it was determined that the NFT system resulted in the greatest profit, neither system was economically viable without factoring in a significant subsidy—50% of the total cost—from the agricultural policies of the European Union. With regards to medicinal plant production, while many studies attempt to maximise secondary metabolite production and yield, or design systems for efficient cultivation, very few examine the cost of production and the value of the produced plants. Dorais et al. [61] calculated the estimated sales value per year of 10 medicinal plants cultivated in an FT system; however, they did not calculate the production costs, so the results are difficult to evaluate. Research into PFALs and similar closed systems is popular due to the ability to optimise environmental conditions, yet a 2014 survey of 165 PFALs in Japan found that 25% were operating at a loss, and 50% were breaking even [102]. Despite this, a PFAL-style system may still be feasible for the production of some aromatic plants due to their advantages over food crops. Plants typically grown in PFALs such as leafy greens need to be picked, transported, and sold quickly before they spoil. On the other hand, medicinal plants are often dried or valuable compounds are extracted from them, extending their shelf life. Bafort et al. [53] determined the feasibility of growing *E. peplus* in a DFT system with LED lighting established in a shipping container. They calculated the income yielded by various levels of processing of the produced biomass. It was determined that the potential gross margin of the production of ingenol mebutate for sale as a laboratory product was between EUR 5000 and 312,000, while the production of a pharmaceutical gel had the potential to generate EUR 160,000 to 650,000 in sales. While not every medicinal plant has the potential to be economically viable when cultivated hydroponically in the current market, the global market of medicinal and aromatic plants is predicted to expand from the current USD 800 million to 50 trillion by 2050 [225]. As the demand for medicinal plants increases, more opportunities for economically viable production are likely to emerge.

## 7. Conclusions

The majority of reports in the literature on the hydroponic cultivation of medicinal plants focus on short-term studies in a controlled environment aimed at identifying the effects of a single environmental factor on the target plant, while papers that focus on scaling up commercial production or on traditional or pharmaceutical use are still few in number. However, they show general trends. The cultivation practices for plants where leaves are used medicinally are generally similar to those of leafy vegetables, including production in PFALs. While trees and shrubs require solid media to support their weight, flowers that grow quickly can be grown in non-circulating liquid culture, as they are likely to be harvested before disease develops, while it is more advantageous to grow longer-lived flowers in pots, allowing them to be spaced out as they grow. Solid culture can be used to cultivate roots or rhizomes, as long as the substrate is sufficiently porous as to allow aeration of the roots. Liquid culture can be used to grow plants with finer roots, while the air-gap system is the most promising with regards to rhizomes. Medicinal fruit and seed crops have been cultivated in pots, similar to their edible counterparts. Some plants also contain medicinally valuable chemicals in plant organs not typically used in traditional medicine, suggesting that their pharmaceutical yield may be greater than initially indicated, potentially increasing the economic viability of hydroponic cultivation. In addition, the incorporation of precursors and forced root permeabilisation may facilitate the extraction of larger quantities of pharmaceutically useful chemicals without the need to harvest the plant. However, there is a dearth of studies focusing on the viability of commercial medicinal plant production, and studies examining the economic feasibility are even rarer. Further examination of existing studies on the commercial viability of hydroponic production could be explored, while experimental studies should also account for the economic viability of their systems.

**Author Contributions:** Designing the review, P.L. and H.R.A.; writing the draft, H.R.A.; supervision, P.L.; editing, H.R.A. and P.L. All authors have read and agreed to the published version of the manuscript.

**Funding:** This research received no external funding.

**Data Availability Statement:** Not applicable.

**Conflicts of Interest:** The authors declare no conflict of interest.

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
