# Peer review of "Hydroponic Cultivation of Medicinal Plants—Plant Organs and Hydroponic Systems: Techniques and Trends"

_horticulturae, doi:10.3390/horticulturae9030349_

Round 1

Reviewer 1 Report

Dear authors, 

Thank you for the opportunity to review your manuscript. 

I have read through the different chapters and have noticed a few mistakes in English language. Minor spell check is required.

The article has gathered a good amount of information and it has been put forward to the reader in a constructive manner. The chapters seem to be complete to an extent and correspond to the principles of a review article. 

Content wise I do not think much else is required, the toic is quite narrow and all of the meaningful articles in this field have been included in this review. 

You sould check the references, they are not all formated accordingly. some journal names are abbreviated some not, some are abbreviated with the dot after the abbreviation some are not and other minor inaccuracies, but its obvious for such an extensive reference list. 

It is always nice to see a figure or two also in review articles, if possible please add some figures on the differences of different hydroponic techniques or plant media etc.

Author Response

Dear Reviewer,

Thank you for your comments.

The references have been re-formatted using EndNote. As we are unsure of the correct abbreviation of some of the journals, we have added the full names, and leave abbreviation of them to the discretion of the editor, as contained within the MDPI guidelines for authors.

With regards to figures, as there are a number of other publications more specifically focused on the hydroponic systems and media, we have chosen to omit figures from this manuscript.

Best Regards

Reviewer 2 Report

1.Introduction.

The introduction includes sufficient background and relevant references.

In the introduction, many topical topics regarding the use and necessity of growing medicinal herbs in closed systems are considered in some detail. The problems of disappearance in nature of some varieties of medicinal herbs due to the high demand for them are considered.

The ways and methods of growing plants in closed systems (NFT, EC, RDT, FT), as well as the problems that arise when growing medicinal plants in closed systems, are considered in detail.

The purpose of the work and the expected result are indicated.

2. Herbs.

The section contains a lot of information and sources on the cultivation of medicinal plants using hydroponics. Unfortunately, the authors did not indicate how effective the recommended growing methods are, for example, compared to growing in the ground (as a traditional technology).

3. Flowers

Many of the questions in this section touched on the effectiveness and commercial application of certain growing methods, however, according to the data presented in this form, it is difficult to understand the magnitude of the expected effect from the use of the recommended technology (in general, this remark applies to all sections of this article).

4. Roots and Rhizomes

This section contains good analytics.

5. Fruits and Seeds

The numbering of sections is broken.

6.Discussion

The discussion is extended, considers most of the issues related to the methods of growing medicinal herbs in hydroponics, perhaps more attention should be paid to the issue of effective commercial cultivation, give real figures for the existing sales volume, priority growing methods and the most popular medicinal crops, and summarize the result in a separate chapter.

7.Conclusion

It is desirable to indicate directions for future research.

Author Response

Dear Reviewer,

Thank you for your suggestions.

The numbering of the later sections has been corrected.

A paragraph has been added to the end of the discussion briefly outlining the issue of commercial cultivation.

A line indicating the direction for future research has also been added to the conclusion.

Best Regards

Reviewer 3 Report

This review investigated the different techniques and trends of hydroponic systems for cultivation of medicinal plants. The topic is very  interesting and well within the aims of the Journal. In general it is quite well written and I think that it can be accepted in the present form. 

Author Response

Dear Reviewer,

Thank you for taking the time to read our manuscript. Attempts to review the grammar and spelling of the manuscript have been made.

Best Regards

Reviewer 4 Report

Comments and Suggestions for Authors

The article presented is a review on the use of different hydroponic systems to cultivate medicinal plants. . There are a variety of hydroponic systems and techniques available, each with its own advantages and disadvantages. As medicinal compounds are often localised within specific plant organs, selecting hydroponic systems that maximise the yield of the desired organs and metabolites is necessary. The aim of this article is to examine and synthesise previous research on the hydroponic cultivation of medicinal plants. While there are previous reviews exploring soilless culture of medicinal species, an examination of the literature through the lens of the hydroponic systems and techniques is absent. However, as of yet there has been no attempt to examine the literature with regards to the cultivation of specific plant organs of medicinal plants in hydroponics and how the choice of system differs depending on the desired plant organ, as well as highlighting studies with novel outcomes that may provide value to the field.

The manuscript is composed according to the requirements of “Horticulturae” for a review preparation. It offers an overview of hydroponic techniques for growing medicinal plants. A summary of applied hydroponic techniques is made according to the type of plant part of the respective species used and could serve as a starting point in any new approach for cultivating any medicinal plant hydroponically.

The following recommendations can be made:

Introduction:

It is unnecessary provide information about the primary metabolites, and secondary metabolites, so it may be ommited. Therefore, the “Introduction” may beggin as followed:

 “Medicinal plant plants, have been used for thousands of years to treat health problems. Nowadays, In China 40% of people rely exclusively on traditional medicines, while in Africa the proportion is even greater at 80% [10].Traditional medicines are of particular importance in rural areas as up to 90% of the world’s rural population use traditional medicine, due to limited access to modern medical facilities [9]. 80% of medicines in traditional Chinese medicine (TCM),  are sourced from plants [8]. Therefore, maintaining the availability of this valuable source of medicinal treatments is of great importance…”

 Conclusion

 The initial part of “Conclusion” : “Medicinal plants are an important medical resource, both for modern in traditional medicine. In order to meet the demand for plant material for traditional use and pharmaceutical extraction, cultivation efforts are required. The use of hydroponic systems to cultivate medicinal plants may be an effective method by which to meet the demands of consumers. While there is an abundance of literature on the topic of hydroponic cultivation of plants, only a minority of papers that focus on medicinal plants. Lower still is the number of papers focusing on expanding commercial production either for traditional or pharmaceutical use—the majority focus on short-term controlled-environment studies, aimed at identifying the effects of a single environmental factor on a target plant.” repeat the written in “Introduction”  and have to be omitted or rewritten

 In continuation, I give one option for the quoted part:

 “The majority of reports in the literature on hydroponic cultivation of medicinal plants focus on short-term studies in a controlled environment aimed at identifying the effects of a single environmental factor on the target plant, while papers that focus on scaling up commercial production or on traditional or pharmaceutical use are still few in number. However, they show a general trend …”

  In conclusion, this manuscript is recommended for publication in “Horticulturae”.

Author Response

Dear Reviewer,

Thank you for your suggestions.

The initial paragraph has been removed and integrated into the second, removing the unnecessary information about primary metabolites. The paragraph now reads as follows:

“Plants produce a variety of different chemicals throughout the course of their growth. Some of these compounds, known asThe secondary metabolites, of some plants  are ben-eficial to human health. They have a wide variety of uses, including reducing inflamma-tion, treating diabetes, reducing cancer risk, preventing cardio-vascular disease, and some have antimicrobial effects [1]. The production of pharmaceutical drugs relies on many of these chemicals. Up to 60% of anti-infectious or anti-tumour drugs available are sourced from nature [2], and approximately 50% of modern pharmaceuticals overall are natural [3]. Currently, there are around 70,000 different species of plants used by both modern and traditional medicinal systems globally [4]. These plants, known as medicinal plants, have been used for thousands of years to treat health problems. In traditional Chinese medicine (TCM), for example, 80% of medicines are sourced from plants [5]. Traditional medicines are of particular importance in rural areas as up to 90% of the world’s rural population use traditional medicine, due to limited access to modern medical facilities [6]. In China 40% of people rely exclusively on traditional medicines, while in Africa the proportion is even greater at 80% [7]. Therefore, maintaining the availability of this valuable source of medicinal treatments is of great importance.”

Your recommendation for the replacement of the initial part of the conclusion has been taken, and the part reads as you suggested.

Best Regards

Author Response

Dear Reviewer,

Thank you for your detailed comments. We have made changed to the following changes to the manuscript resulting from your suggestions.

Page 1, Line 24-25: The key word “hydroponics” has been added.

Page 1, Line 30-32: The line has been removed at the suggestion of another reviewer.

Page 2, Line 61-62: The sentence has been rephrased to: Consequently, it is key that the environmental impact of using medicinal plants is reduced, and that the safety and quality of medicinal material is maintained.

Page 2, Line 68-71: The following text has been added:

For example, in a review of studies examining the cultivation of Hypericum perforatum, Bruni and Sacchetti [14] observed that cultivation studies conducted in different countries reported hypericin, pseudohypericin and hyperforin concentrations that varied widely. Similarly, differences in salidroside yield of Rhodiola sachalinensis roots were observed in response to soil characteristics, such as organic matter, nitrogen, phosphorous and potas-sium content, as well as pH  [15]. Additionally, seasonal variation in temperature and moisture have also been attributed to differences in the levels of phenolic compounds and saponins in Tulbaghia violacea, Hypoxis hemerocallidea, Drimia robusta and Merwilla plumbea plants [16]. Continuous cropping of some species can also result in declining yields over time. This has been observed in Panax ginseng, whose yields decline due to the prolifera-tion of soilborne disease attributed to the accumulation of root [17].

Page 2, Line 88-92: Abbreviations added.

Page 3, Line 114-143: Paragraph has been re-organised to:

Yoshimatsu [26] observed an increase in both biomass accumulation and glycyrrhizin content of Glycyrrhiza uralensis in hydroponics, Surendran et al. [27] recorded that the Mentha spicata yield, organic acid concentration, and antioxidant content was higher in hydroponically-grown plants, and Duan et al. [28] found that Trichosanthes kirilowii had greater plant height and number of leaves per plant, as well as a 100% seedling survival rate in hydroponics, compared with an 87% survival rate in soil. Conversely, Afreen et al. [29] found that hydroponic cultivation resulted in a reduction of root and shoot growth of G. uralensis, while Souret and Weathers [30]observed a lower increase in fresh weight in Crocus sativus when it was grown in hydroponics. Similarly, Maggini et al. [31] found that the echinacoside content of hydroponically grown Echinacea angustifolia was below the 1% standard required by the European Pharmacopeia, and Strzemski et al. [32] remarked that cultivation of Carlina acaulis in hydroponics resulted in lower carlina oxide content than when it was grown in soil.

Table 1: Spelling Corrected. Names removed.

Page 6, Line 186: Spelling Corrected

Page 6, Line 215-217: Line changed to: The aim of PFALs isaim to optimise environmental conditions to maximise growth and phytonutrient content of high-value plants like vegetables and medicinal plants [101].

Table 2: Spelling Corrected

Page 9, Line 271-272: Sentences merged.

Page 9, Line 280-281: Sentence rephrased to:

Cultivation of Dandelion (T. officinale) plants using either NFT or DFT found that root and shoot yield was greater when DFT was used.

Page 9, Line 295-298: Word Added.

Page 9, Line 298-302: Sentence rephrased to:

However, the initial observations of Sellar [99] were that soil-grown and soilless D. stramonium plants were initially comparable in growth, and that the hydroponically cultured plants showed greater yield upon maturation. Conversely, other studies have shown that initial corm size is a determining factor on the yield of saffron [127,150,151], which may suggest that the yield of some flower species may be determined by their state prior to transplantation into hydroponics

Page 10, Line 334-337: Rephrased to:

The authors proposed that mixture results in a medium with a high water-holding capacity, while also maintaining a high air-filled porosity—supplying plants with both water and nutrients while also aerating the roots of the plant.

Page 12, Line 390-392: Statement amended to:

However, despite these results, the plants shown in the figures of both Tabatabaei [167] and Thakur et al. [171] show reduced rhizome development, and Dorais et al. [60] makes no mention of the presence of rhizomes, suggesting that while root growth may be improved, growth of the medicinally-valuable rhizome is reduced.

Hope this helps clarify the intended meaning.

Page 13, Line 436-437: Spelling corrected.

Page 14, Line 455: Vestigial reference deleted.

Page 15, Line 474-475: Spelling corrected

Page 17-18, Line 618-621: Line replaced with:

Some plants also contain medicinally valuable chemicals in plant organs not typically used in traditional medicine, suggesting that their pharmaceutical yield may be greater than initially indicated, potentially increasing the economic viability of hydroponic cultivation. 

Best Regards